# Cancer Diagnosis by Neural Network Analysis of Data from Semiconductor Sensors

**DOI:** 10.3390/diagnostics10090677

**Published:** 2020-09-05

**Authors:** Vladimir I. Chernov, Evgeniy L. Choynzonov, Denis E. Kulbakin, Elena V. Obkhodskaya, Artem V. Obkhodskiy, Aleksandr S. Popov, Victor I. Sachkov, Anna S. Sachkova

**Affiliations:** 1Tomsk National Research Medical Center of the Russian Academy of Sciences, Cancer Research Institute, 5 Kooperativny Street, 634009 Tomsk, Russia; chernov@tnimc.ru (V.I.C.); choynzonov@tnimc.ru (E.L.C.); kulbakin_d@mail.ru (D.E.K.); 2Laboratory of Chemical Technologies, National Research Tomsk State University, 36 Lenin Avenue, 634050 Tomsk, Russia; lenaobx@yandex.ru (E.V.O.); asptomsktpu@gmail.com (A.S.P.); 3School of Nuclear Science & Engineering, National Research Tomsk Polytechnic University, 30 Lenin Avenue, 634050 Tomsk, Russia; asachkova@tpu.ru

**Keywords:** malignant neoplasm, classification, electronic nose, neural network, metal oxide semiconductor sensor, gas analyzer

## Abstract

“Electronic nose” technology, including technical and software tools to analyze gas mixtures, is promising regarding the diagnosis of malignant neoplasms. This paper presents the research results of breath samples analysis from 59 people, including patients with a confirmed diagnosis of respiratory tract cancer. The research was carried out using a gas analytical system including a sampling device with 14 metal oxide sensors and a computer for data analysis. After digitization and preprocessing, the data were analyzed by a neural network with perceptron architecture. As a result, the accuracy of determining oncological disease was 81.85%, the sensitivity was 90.73%, and the specificity was 61.39%.

## 1. Introduction

Most of the methods of early diagnosis of malignant neoplasms are invasive, operator-dependent and expensive. In this regard, in a number of cases such methods are not available for the majority of the population and may not match the principles of modern screening for malignant neoplasms [1].

The modern recommended strategy for examining patients with head and neck tumors is based on the use of endoscopic, radiological and morphological diagnostic methods [2]. The effectiveness of these diagnostic methods also depends on the experience of the polyclinic doctor and the availability of instrumental and laboratory diagnostic methods. 

Treatment of patients with locally advanced malignant tumors of the oropharyngeal region and larynx is often associated with a whole complex of negative consequences: disability, impaired physiological functions, severe cosmetic losses and the occurrence of psycho-emotional trauma [3]. This circumstance directs clinical oncologists and specialists in related fields to search for affordable and effective methods for diagnosing malignant tumors at early stages, which reflects modern principles of screening methods for diagnosing tumors of the oropharyngeal region and larynx - reproducibility, low cost, independence from the human factor (reduction of error), and the possibility of use in non-specialized clinics by primary care physicians (therapists, ENT doctors, dentists) [4].

Analysis of breath samples is a promising method for population screening. Gas chromatography–mass spectrometry is a widely used method for this analysis. Many works demonstrate the effectiveness of this method in diagnosing tumor pathology [5,6,7,8,9,10,11,12]. However, it has some disadvantages, e.g., implementation requires highly qualified personnel, the method does not have sufficient mobility and it requires relatively expensive equipment [13,14]. Moreover, a relatively large amount of time is needed, not only for the analysis itself, but also regarding its interpretation.

“Electronic nose” technology is a noninvasive method for diagnosing malignant neoplasms, which involves a set of gas sensors and the ability to use a specific method of information processing [15]. This method does not have the disadvantages of the above. The necessary equipment is relatively inexpensive and the method has high mobility due to its small size. Highly qualified personnel are not required since probability is determined automatically using a neural network without human intervention, and the technique is fast due to the ability of the neural networks to process information at high speed and the ability of gas sensors to provide data without delay. In this study, the “electronic nose” was used to identify volatile organic compounds (VOCs) in the exhaled air of patients with malignant neoplasms and of healthy people. The obtained data were structured in order to establish whether the sample belonged to a healthy person or not. In human exhaled air more than 3000 VOCs have been found that can be differentiated as endogenous or exogenous. Endogenous VOCs, which originate from internal metabolic processes, carry the most important information about the state of the individual’s health. However, the number of such particles in exhaled air is extremely small, i.e., 1–5000 units per billion [16,17]. The concentration can also vary considerably based on the lifestyle conditions of individuals, their weight, age, gender, bad habits, the presence of various diseases, etc. [18,19,20,21,22]. Exogenous VOCs, which result from environmental changes, can interfere with research, but they carry valuable information about a person’s habits (for example, smoking). Previously in work by Phillips et al. [23], it was shown that lung cancer can change the percentage of some VOCs present in human breath. These substances can be considered as markers of this disease.

Machine learning enables qualitative and quantitative identification of chemical substances based on large amounts of data. Based on these data, “electronic nose” systems use the following approaches [24]:Support vector machine (SVM);k-nearest neighbors algorithm (k-NN);Artificial neural network (ANN).

Most of the works related to the analysis of VOCs and their relationship with oncological diseases use ANN. Analysis of previous research [15,16,17,18,19,20,21,22,23,24,25] has shown that the breath samples can be taken in several ways. For instance, Schmekel et al. [25] used a thick foil, similar to the aluminum foil used to wrap food. This foil was folded in two layers so that outside gases could not penetrate the material. The patients had refrained from smoking and had not eaten for two hours before taking the sample by a simple, slow exhalation. This foil held three liters of exhaled air. In some cases, Tedlar bags were used [26]. In other works [27,28,29], prolonged breath was exhaled directly into the chamber, which included sensors. In [28], inhalation was performed through a charcoal filter and exhalation was performed into the sampling chamber. Afterwards, the system did not collect data for two minutes, thereby minimizing the influence of the environment. The data were then collected for three minutes.

In another study [29], patients breathed into the device for five minutes while holding their nose and using a special outlet to prevent bacteria, viruses and airborne debris from entering the device chamber. Kesavaraj and Sukumaran [26] used Tedlar bags. Before each sample analysis, the chamber was purged using the sensors, with dry air for 100 s, then air was pumped into the chamber from the bag for 40 s and analyzed for 30 s. The samples were finally removed from the chamber within 140 s. In a study by Herman-Saffar et al., [27] patients breathed through a mask connected to an electronic nose for 40 s. Schmekel et al. [25] injected samples within 40 s, and the average value of the last two measurements was used for calculations. In another study [28], patients breathed through an “electronic nose” device for five minutes (two minutes for preparation and three minutes for data acquisition). The composition of the air was measured every 20 s using two 32-step sinusoidal temperature modulations on the sensor surface. The measured results consisted of vectors of 64 values every 20 s for each of the three sensors. Previous studies, with varying degrees of success, have used the “electronic nose” device to diagnose lung [15,17,25,26,29,30,31,32,33,34,35], colon [36,37], head and neck [34,37,38], urinary bladder [37], prostate [39], stomach [28] and breast cancer [27].

Blatt et al. [17] researched the use of various classification methods to study lung cancer in 101 people (58 healthy individuals, 43 patients with lung cancer). They analyzed the final part of the exhalation, since, according to the authors this contains the most valuable information. Classification was carried out by such methods as modified k-NN, classifiers based on linear and quadratic discriminant functions, and ANN. The accuracy rate of determining healthy people with ANN was 88.8% (the highest, in comparison to other works), and 93.8% when identifying patients. The authors concluded that a higher percentage of false negative results was more important than the highest accuracy, which did not differ significantly from the rest; therefore, they concluded that primacy could be assigned to the modified k-NN method (88.2% and 96.3%, respectively) and the quadratic discriminant classifier (88.2% and 96.3%, respectively). Perhaps an increase in the number of learning epochs in ANN might significantly enhance the accuracy. In the above mentioned work the number of learning epochs was 20, which might not be enough considering the small amount of statistical data. In addition, a single-layer neural network was used, which was not sufficient, given the specificity of the data. Despite these shortcomings, accuracy turned out to be quite high. 

It is worth highlighting the work of Waltman and van Roermund [39], in which the commercially available “electronic nose” Aeonose was used to diagnose various diseases, including malignant neoplasms. The device consists of three metal oxide sensors with different characteristics of the sensing element surface. The study analyzed the exhaled air of 85 patients, 32 of whom had prostate cancer with benign prostatic hyperplasia, the remaining 30 being healthy. The patients breathed slowly for over five minutes using the device and air was inhaled through a charcoal filter. Along with all the necessary procedures, from the beginning of one measurement to the beginning of the next no more than 15 min passed, including direct measurement, preliminary data processing and preparation for the next sample. The functionality of the method was assessed by cross-validation. For the training of the neural network, 90% were evenly selected from the entire group. The remaining 10%, who did not participate in training, were classified by received ANN. The recognition accuracy was 77%, which is comparable to studies of markers in the blood (67–83%) and primary biopsies (75–80%). The same device was used in many other works [28,29,34,37,40]. The Cyranose 320 “electronic nose” is a more sophisticated, compact, and mobile solution, consisting of an array of 32 semiconductor sensors capable of analyzing many gases, not limited to the medical field. Comparative analysis of exhalation using the “electronic nose” Cyranose 320, and urine using gas chromatography mass spectroscopy was used by Herman-Saffar et al. [27]. For respiration analysis, 48 patients with breast cancer and 45 healthy individuals were recruited, 37 and 36 individuals were recruited for urine analysis, respectively. All data were analyzed by ANN. As a result, the accuracy of the analysis using the electronic nose was, on average, 95.2%, which significantly exceeded the urine analysis study. No work exists in the literature regarding the use of other commercially available electronic nose devices for diagnosing cancer with comparable accuracy.

The results obtained by Wang et al. [31] are scientifically interesting. Researchers used a hybrid scheme of metal oxide sensors (MOS) and surface acoustic wave sensors (SAW). The hybridity was justified by higher sensitivity to more VOCs. The MOS sensors detected low-molecular-weight VOCs and the SAW detected high-molecular-weight VOCs. The device consisted of nine MOS and one SAW. Analysis of the preprocessed data was carried out by linear discriminant analysis, partial least squares, and ANN. Operability was assessed by cross-validation. As a result, analysis using ANN gave the best result, with a sensitivity of 93.62% and a selectivity of 83.37%. The authors used three different ANNs: for MOS, for SAW sensors, and to summarize the results.

In the work of Ozsandikcioglu et al. [26], the abovementioned methods were used to classify air samples taken from 18 people, five of whom were healthy and the rest diagnosed with lung cancer. When using a hybrid system of MOS and quartz crystal microbalance (QCM) sensors, an accuracy of 91.4% was shown for the nearest neighbors method, 85.7% was shown for the support vector machine, and 91.4% was obtained for the ANN.

Schmekel et al. [25] divided patients into three groups: 1. who had lived more than a year since testing; 2. who had died during the year after testing; and 3. healthy individuals. Before testing, 17 patients had stage four lung cancer and six had stage three. Authors managed not only to find a correlation between sick and healthy subjects, but also to predict approximately the time of death. Thus, with sufficient statistics, the “electronic nose” was also proven to determine the survival rate of patients. From the above, it was concluded that metal oxide sensors and ANN can be useful in electronic nose devices for the diagnosis of oncological diseases.

## 2. Materials and Methods

In this work, breathing samples were taken from 59 people aged 22 to 71 years. All subjects were divided into two groups; a control group without confirmed malignancy, and patients with morphologically verified malignancies of the oral cavity, oropharynx, larynx, tongue, or lung, phases T_1–4_T_0–3_M_0–1_. Information on the samples taken from the study group is presented in Table 1.

All subjects in the study group underwent a standard medical screening to clarify the stage and extent of the tumor process. For this, endoscopic (Evis Exera II Olympus CV-180, Tokyo, Japan) and X-ray diagnostic (Siemens Magnetron Essenza 1.5 T and Siemens Somatom Emotion 6, Heusenstamm, Germany) methods were used, as well as morphological verification of the biopsy material. 

The study group with malignant pathology totaled 36, 6 women and 30 men. By age category, two people were in the range of 20–40 years old and 34 people were aged 40–75 years old.

The control group included persons with no data on malignant pathology at the time of the study (according to anamnestic information and data from a previous examination). The criteria for exclusion from the control group included history of malignancy, any treatment for malignant tumor pathology, less than 18 years old, infectious disease in the acute phase, antibiotic treatment, pregnancy, and breastfeeding. This group comprised 16 women and 7 men. Among the patients in this group, 14 were in the 20–40 age group and 9 were in the 40–50 age group.

Air sampling of each person was carried out through a 5-L two-layer bag (NIKI-MLT, Saint-Petersburg, Russia). The outer layer was made of Ethylene-Vinyl alcohol (EVOH) material 90 microns thick and the inner layer was made of Very Low-Density Polyethylene (VLDPE) 50 microns thick. The time interval from the moment of sampling to its processing by the device did not exceed 12 h.

Prior to collection of breath samples, patients abstained from food and drink (other than water), they did not use any personal hygiene products such as scented soaps or perfumes, and refrained from smoking and brushing their teeth for at least six hours before the study. Participants did not consume alcohol for at least 24 h before the study. Thus, the most optimal time to collect samples was in the morning. In the study group, samples of exhaled air were taken after all diagnostic procedures. The study was approved by the Bioethical Committee of the Cancer Research Institute, Tomsk National Research Medical Center of the Russian Academy of Sciences (Order on creation No. 57-p dated 23 December 2010).

A gas analysis system to analyze the patient’s breath samples was developed (Figure 1). This method can be used either for taking an air sample directly or by using a sample bag. Since this work was carried out in the context of the coronavirus pandemic, we used the bag sampling mode. 

The system (Figure 1) consists of a sampling chamber (5) (quartz glass and brass flanges), inside which is a module containing 14 MOS sensors (Table 2), 6 constantly running fans distributing sample air evenly throughout the chamber, and a MEGA2560 PRO control board. Before the sensors were installed, they were preheated for 200 h. After installation, a load resistor was selected for each sensor, providing a signal level in the range of the analog to digital converter scale.

The indoor module was connected with a data and power cable to the interface module (6). The interface module contains I2C port expanders, through which switches and control buttons are implemented. With the help of these, an 8-relay block is connected, providing control of 7 valves (2) and a pump (4). The interface module is connected to the data analysis computer via a USB interface. Additionally, an RS-232 interface with a DB-9 connector was implemented.

The sampling device was powered by two sources, a 5 V source and a 24 V source (1). Air to purge the sampling chamber was supplied by a membrane pump (4) and passed through a filter filled with a mixture of zeolite and silica gel (3).

Figure 2 shows the generalized structure of the gas analytical system for the bag sampling mode. The bottom valve (2) is for the timely supply of the gas sample. The upper two valves (1, 3) are used to purge the chamber with a pump. The sensors operate in a thermal cycling mode of 3.5 s of heating and 5.5 s of cooling. Prior to sampling, the sensors were warmed up until the readings became stable, which took from 30 min to 3 h depending on the instrument idling time. While preparing the gas analytical system for measurements, the pump blew clean air through the sampling chamber. When the readings of the sensors stabilized, an air sample was injected from the bag allowing an array of numerical values to be obtained. The data are shown in Figure 3.

Collecting data for a gas sample includes several stages:A half-filled sample bag is connected to the inlet valve. The bag is loaded with the same weight for all measurements. At this moment, the inlet (2) and outlet (3) valves of the sampling chamber are closed.In the interface of the data collection program, a button is pressed and the reading and transmission of data at a frequency of 30 Hz to a personal computer begins.At the moment of transition from the heating phase to the cooling phase of the sensors (Figure 3—mark 5000 ms), valves (2) and (3) open for exactly one second. The same opening time is used for valves (2) and (3), and the weight of the load on the bag provides the same volume of gas sample (~ 250 mL) introduced into the 1 L sampling chamber.After the valves are closed, data collection continues with the sample gas inside the chamber. The process of collecting data continues until the time stamp of 90,000 ms. The total residence time of the sample in the chamber does not exceed 90 s.After 90 s, the sample chamber purge is automatically turned on for 120 s. The instrument is then ready to analyze the next sample. These time intervals were selected experimentally and take into account the inertia of the sensors.

In the measuring module software was established so that the data obtained from channels 15 and 16 (without installed gas sensors) exhibited given constant values that were different from zero during heating, but null values during cooling (signals 15 and 16 are in the form of a meander in the graph). This is done to accurately separate the thermal cycling periods in the preprocessing stage, i.e., to synchronize the measurements. As a result, a record of 43,200 values is saved in the database for each individual patient (14 sensors and 2 channels, each 30 Hz × 90 s = 2700). Data acquisition and analysis software are implemented in the Qt Creator development environment. The neural network processing program was developed in the Nvidia Nsight Eclipse Edition environment using the cudaNN library. The neural network was trained and tested using a GeForce GTX 1070 graphics accelerator.

## 3. Results

In this work the diagnosis of oncological diseases is a classification task, and that is why the perceptron was chosen as the architecture of the neural network. The ratios of 4 and 1 periods of thermal cycling of all 14 sensors, from the heating-cooling point to the cooling-heating point (5.5 s × 30 Hz = 165 values), were fed to the input of the neural network. In the course of the work, it was experimentally found that, if we take every 10th value, then the error does not increase, thus the size of the input data array significantly reduces (from 165 values for each sensor to 16). Consequently, 16 × 14 values obtained from gas sensors, such as age, sex and smoking, were fed to the input of the neural network. The total dimension of the input layer was 227 values. The hidden layer has dimension input × 2 or 454 values. The output layer has one neuron that receives “0” in the absence of cancer and “1” if it is present. The transfer function of sigmoid neurons was used on all layers of the neural network.

The error in solving the neural network classification problem was calculated by cross-validation. 59 datasets were randomly divided into 5 groups. In 4 groups were 12 sets, in the 5th were 11. Then training was carried out in 4 groups, using a test set that did not participate in training, based on the fifth group. As a result, 5 experiments were obtained, for each of which the parameters of the informativeness of the investigated diagnostic method were calculated according to the methods [17,41]. The division into groups was randomly performed 20 times, due to the small number of data sets. For each group five experiments were carried out by it as described above. Table 3 shows the average values for 100 experiments (20 × 5) for the developed gas analytical system.

Taking into account the small number of data sets, in order to exclude overfitting of the neural network, the control parameter for stopping training was chosen for accuracy in determining the test set. The stop was made at the moment when the error in determining the training sets decreased, but the error in determining the test set began to grow. This point was observed between 500 and 8000 learning epochs.

As a result, a complex was developed, consisting of a technique, a device and software, capable of analyzing gas samples in two modes - direct breathing into the chamber, or the use of bags for gas samples. It was possible to use only remote bag sampling due to the COVID-19 pandemic. The database of patients recruited provides automatic search for certain attributes, for operational retraining, or for analyzing a patient with a variation in input data. The metadata of the database includes data on age, sex, smoking, type and location of the tumor, stage of the disease, presence of metastases, undergoing treatment at the time of sampling, the presence of diabetes mellitus, and an array of data on the sample obtained. The database can be used in the future to conduct experiments to improve the accuracy of the neural network.

## 4. Conclusions

In this study, a set of data on exhaled air was analyzed for two groups of patients, the study group of 36 people and the control group of 23 people. The nature of pathology in the study group was varied and included various types of localization of malignant neoplasms of the respiratory tract, including lung, larynx, oral cavity, oropharynx and tongue. The number of samples of exhaled air in patients with various types of pathology according to age group was also different. The use of such experimental data ensured achievement of the parameters of the gas analytical system at a specificity level of 61.39% and a sensitivity of 90.73%. 

The obtained indicators of specificity and sensitivity were comparable with modern high-precision X-ray methods for diagnosing tumors of the respiratory tract (MRI, CT, PET). According to various sources, the specificity of PET with 18F-FDG and MRI in the diagnosis of tumor lesions in the head and neck region ranges from 43% to 95%, with sensitivity ranging from 64% to 92% [42]. These values are very similar to those obtained using PET/CT and CT in the diagnosis of lung cancer, where computed tomography has a sensitivity of 72.1% and a specificity of 90.3%, and PET/CT has a sensitivity of 90.1% and a specificity of 96.2% [43].

Modern X-ray methods, it should be noted, are financially expensive due to their use of expensive equipment, which is often only available in large clinical centers. In contrast, the developed gas analytical system does not require the use of expensive equipment and consumables and is also quite mobile and can be used in a medical institution of any level. 

## Figures and Tables

**Figure 1 diagnostics-10-00677-f001:**
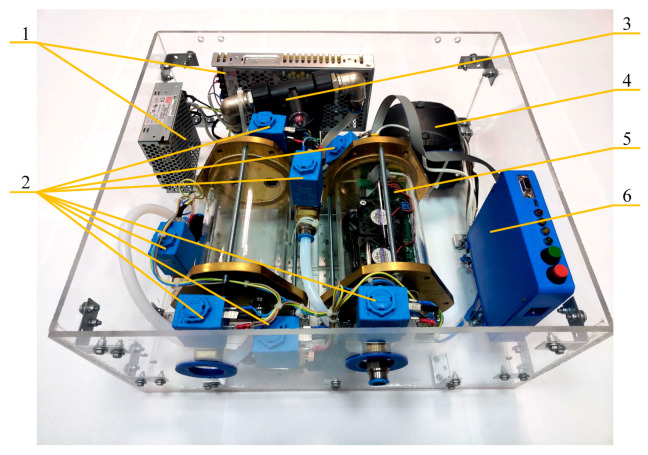
View of the sampling device as part of the gas analytical system.

**Figure 2 diagnostics-10-00677-f002:**
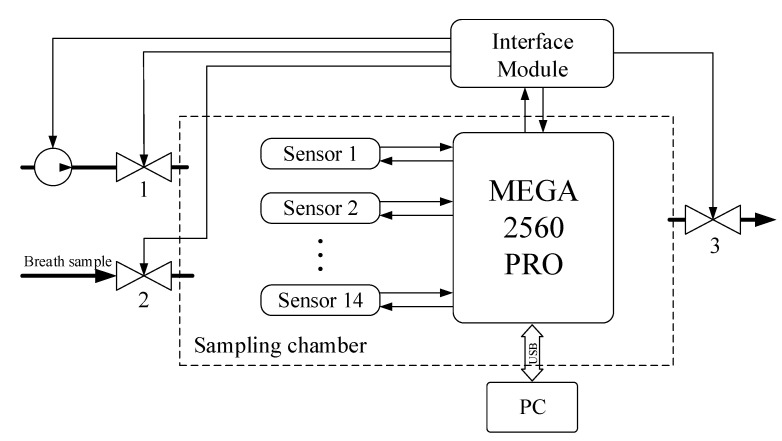
Generalized structure of the gas analytical system. Bold black arrows—gas flow; regular black arrows—control signal line; white circle—pump; dotted line box—sampling chamber; 1, 2 and 3—valves.

**Figure 3 diagnostics-10-00677-f003:**
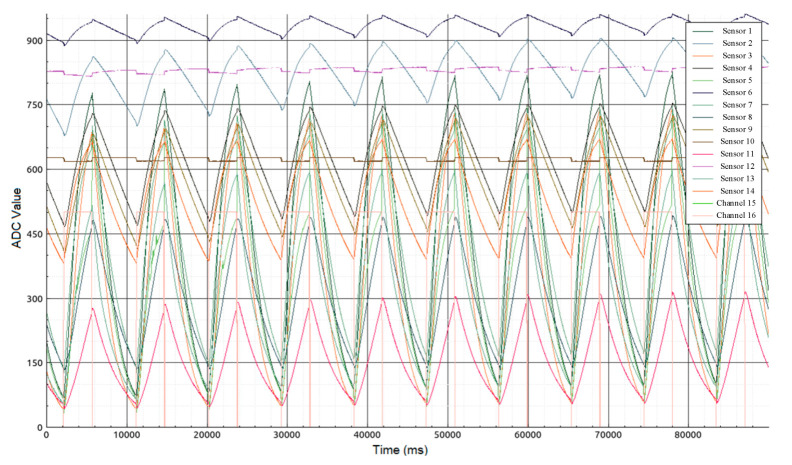
Form of sensor signals (ADC—analog-to-digital converter).

**Table 1 diagnostics-10-00677-t001:** Characteristics of the study set of patients.

Stage of the Pathological Process (According to TNM System)	Number of Patients with Malignant Pathology by Tumor Location					
	Morphological Diagnosis Confirmation Method					
	Lungs	Larynx	Oral Cavity	Oropharynx	Tongue	The Mucous Membrane of the Alveolar Process of the Lower Jaw
Stage T_1_	2			1		
Stage T_2_	3	2	3	2	4	1
Stage T_3_	9	1		2		
Stage T_4_	4		1	1		

TNM System—TNM Classification of Malignant Tumors.

**Table 2 diagnostics-10-00677-t002:** Sensors as part of a gas analytical system.

№	Sensor	Sensitivity
1	MP503	Alcohol, Smoke, Isobutane, Methanol
2	WSP2110	Toluene, Benzene, Methane
3	MQ3	Alcohol
4	MQ2	Isobutane, Propane, Methane, Alcohol, Hydrogen, Smoke
5	MQ7	CO
6	MQ131	O_3_
7	MQ135	NH_3_,NO_x_, Alcohol, Benzene, Smoke, CO_2_
8	MQ8	Hydrogen
9	MQ138	n-Hexane, Benzene, NH_3_, Alcohol, Smoke, CO
10	TGS822	Methane, CO, Isobutane, n-Hexane, Benzene, Ethanol, Acetone
11	TGS2602	Ethanol, Toluene, NH_3_, H_2_S
12	TGS2620	Methane, CO, Isobutane, Hydrogen, Ethanol
13	TGS2600	Isobutane, Hydrogen, Ethanol
14	TGS2603	Hydrogen, H_2_S, Ethanol, Methanethiol, Trimethylamine etc.

Sensors N1–9 (Winsen Electronics Technology Co., Ltd., Zhengzhou, China); Sensors N10–14 (FIGARO ENGINEERING INC., Osaka, Japan).

**Table 3 diagnostics-10-00677-t003:** Achieved parameters of the gas analytical system.

Parameter	Value
Accuracy	81.8%
Sensitivity	90.7%
Specificity	61.4%

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
