# Peer review of "Cancer Diagnosis by Neural Network Analysis of Data from Semiconductor Sensors"

_diagnostics, 2020, doi:10.3390/diagnostics10090677_

Round 1

Reviewer 1 Report

This manuscript has good thorough descriptions of methods used, but still has significant limitations that require revisions as follows:

Introduction

  1. Update older references with newer ones on the same points
  2. A number of important concepts relating to this subject [electronic-nose related clinical diagnostics] have not been covered in the Introduction including the following important topics [not mentioned] which should serve as keywords for more recent literature searches including: 1) biomedical applications, 2) clinical disease diagnostics, 3) cancer biomarker metabolites, 4) noninvasive early disease diagnosis, 5) electronic-nose technologies, 6) early disease detection, and 7) clinical practice.
  3. Avoid beginning a new paragraph with a first sentence starting with a preposition or independent clause. Instead, use a direct statement with the preposition or clause at the end of the sentence. Global edit throughout the manuscript to correct this problem.
  4. paragraphs 6 and 15 - single-sentence paragraph; combine with other paragraph or expand
  5. paragraph 9, add: by Blatt et al. [14]
  6. paragraph 10, add: by Waltman and Roermund [38]
  7. paragraph 11, add: by Herman-Saffar et al. [26]
  8. paragraph 12, add: Li et al. [30]
  9. paragraph 13, add: Ozsandikcioglu et al. [25]
  10. paragraph 14, add: by Schmekel et al. [24]

Materials and Methods

Table 1 - move 'Morphological diagnosis confirmation method' (heading) to above the 5 tumor location subheadings (Lungs, Larnyx etc.) with a line drawn above these and below heading

Results - very lean

Provide detailed data tables of e-nose results for the 5 subheading tumor locations (taken from patients in each category) as listed in Table 1 with appropriate statistics to show significant differences. Also, include appropriate and necessary controls

Discussion

Conclusions are not supported due to insufficient e-nose data to demonstrate the basis of conclusions

References

No publication year was provided for references 6, 7, and 8

Author Response

Reply to Reviewer #1

We are thankful to Reviewer for benevolent and positive reply and useful comments. We’ve made all the corrections suggested by Reviewer.

  • Update older references with newer ones on the same points

Reply:  We removed these references, they were additional (Taucher, J et al. 1997: Gordon, S.M . et al. 1985)

2  A number of important concepts relating to this subject [electronic-nose related clinical diagnostics] have not been covered in the Introduction including the following important topics [not mentioned] which should serve as keywords for more recent literature searches including: 1) biomedical applications, 2) clinical disease diagnostics, 3) cancer biomarker metabolites, 4) noninvasive early disease diagnosis, 5) electronic-nose technologies, 6) early disease detection, and 7) clinical practice.

Reply: We added the following information:

Page 2; Lines 28-44:

Most of the methods of early diagnosis of malignant neoplasms are invasive, operator-dependent and expensive. In this regard, in a number of cases such methods are not available for the majority of the population and may not match the principles of modern screening for malignant neoplasms [1].

The modern recommended strategy for examining patients with head and neck tumors is based on the use of endoscopic, radiological and morphological diagnostic methods [2]. The effectiveness of these diagnostic methods also depends on the experience of the polyclinic doctor, the availability of instrumental and laboratory diagnostic methods.

Treatment of patients with locally advanced malignant tumors of the oropharyngeal region and larynx is often associated with a whole complex of negative consequences: disability, impaired physiological functions, severe cosmetic losses, and the occurrence of psycho-emotional trauma [3]. This circumstance directs clinical oncologists and specialists in related fields to search for affordable and effective methods for diagnosing malignant tumors at early stages, which reflects the modern principles of screening methods for diagnosing tumors of the oropharyngeal region and larynx - reproducibility, low cost, independence from the human factor (reduction of error), and also the possibility of using in non-specialized clinics by primary care physicians (therapists, ENT doctors, dentists) [4].

Also we added 3 new references: (Page 9; Lines 323-329):

  1. Shah, J.; Patel, S.; Singh, Bh.; Wong, R. Head and Neck Surgery and Oncology, 5th ed.; Elsevier Science Publishing Company, Inc.: Netherlands, 2019;  p. 859, ISBN 9780323417907.
  2. The National Comprehensive Cancer Network. Clinical Practice Guidelines in Oncology. Available online: https://www.nccn.org/professionals/physician_gls/default.aspx.
  3. Harris, A.; Lyu, L.; Wasserman-Winko, T.; George, S.; Johnson, J. T.; Nilsen, M. L. Neck Disability and Swallowing Function in Posttreatment Head and Neck Cancer Patients. Otolaryngology – Head and Neck Surgery 2020, 1-8.

3     Avoid beginning a new paragraph with a first sentence starting with a preposition or independent clause. Instead, use a direct statement with the preposition or clause at the end of the sentence. Global edit throughout the manuscript to correct this problem.

Reply:   Corrected

4  Line 78: Paragraph 6 is combined with paragraph 7. Line 161: Paragraph 15 is combined with paragraph 14.

5     paragraph 9, Line 104: added: by Blatt et al. [14]

6     paragraph 10, Line 119: added: by Waltman and Roermund [38]

7     paragraph 11, Line 135: added: by Herman-Saffar et al. [26]

8     paragraph 12, Line 142: added: by Wang et.al. [30]

9     paragraph 13, Line 151: added: Ozsandikcioglu et al. [25]

  1. paragraph 14, Line 156: added: by Schmekel et al. [24]

Reply:   Corrected

11  Materials and Methods.

Table 1 - move 'Morphological diagnosis confirmation method' (heading) to above the 5 tumor location subheadings (Lungs, Larnyx etc.) with a line drawn above these and below heading

Reply:   Corrected. Table 1: Line 171.

12  Results - very lean

Reply:  We corrected this section and added the following information:

  • Lines 254 -272:

The ratios of 4 and 1 periods of thermal cycling of all 14 sensors from the heating-cooling point to the cooling-heating point (5.5 seconds * 30 Hz = 165 values) were fed to the input of the neural network. In the course of work, it was experimentally found, if we take every 10 value, then the error does not increase, thus, the size of the input data array significantly reduces (from 165 values ​​for each sensor to 16). Consequently, 16 * 14 values ​​obtained from gas sensors, such as age, sex and the fact of smoking, were fed to the input of the neural network. The total dimension of the input layer was 227 values. The hidden layer has dimension input * 2 or 454 values. The output layer has one neuron that receives "0" in the absence of cancer and "1" if it is present. The transfer function of sigmoid neurons was used on all layers of the neural network.

The error in solving the neural network classification problem was calculated by cross-validation. 59 datasets were randomly divided into 5 groups. In 4 groups were 12 sets, in the 5th - 11. Then training was carried out in 4 groups, using a test set that did not participate in training, based on the fifth group. As a result, 5 experiments were obtained, for each of which the parameters of the informativeness of the investigated diagnostic method were calculated according to the methods [17, 41]. The division into groups was randomly performed 20 times, due to the small number of data sets. For each group five experiments were carried out by it as described above. Table 3 shows the average values ​​for 100 experiments (20 * 5) for the developed gas analytical system.

  • Lines 275 -288:

Taking into account the small number of data sets, in order to exclude overfitting of the neural network, the control parameter for stopping training was chosen the accuracy in determining the test set. The stop was made at the moment when the error in determining the training sets decreased, but the error in determining the test set began to grow. This point was observed between 500 and 8000 learning epochs.

As a result, a complex was developed, consisting of a technique, a device and software, capable of analyzing gas samples in two modes - direct breathing into the chamber, or the use of bags for gas samples. It was possible to use only remote bag sampling due to the COVID-19 pandemic. The database of patients was recruited, which provides automatic search for certain attributes for operational retraining, or for analyzing a patient with a variation in input data. The metadata of the database includes attributes with data on age, sex, smoking, type and location of the tumor, stage of the disease, presence of metastases, undergoing treatment at the time of sampling, the presence of diabetes mellitus, as well as an array of data on the sample obtained. The database can be used in the future to conduct experiments to improve the accuracy of the neural network.

13  Provide detailed data tables of e-nose results for the 5 subheading tumor locations (taken from patients in each category) as listed in Table 1 with appropriate statistics to show significant differences. Also, include appropriate and necessary controls.

Reply: 

In the present study, we did not determine the localization of a malignant neoplasm. The aim was to determine the presence or absence of it. We think that this idea of determining the type of cancer by an electronic nose and a neural network is a very interesting and promising. Further research are going to determine this possibility. All our attempts to visually distinguish the cumulative picture of signals from all 14 sensors at different localizations of the neoplasm were unsuccessful. This is provided only by a neural network, while it considers not one separate sensor, but a set of 14 for each individual patient and recognizes the general picture (pattern) characteristic of a cancer patient. It should be noted that the investigated technique does not claim to replace completely  the existing traditional approaches. This technique is a useful complement to them. It can be used for early non-invasive diagnosis of malignant neoplasms, for example, during an annual clinical examination in medical institutions of population. After diagnosis it could be recommended for patient  to undergo a more detailed examination using diagnostic methods, that have long been included in medical practice, such as PET / CT, CT and morphological analysis. As a result, the likelihood of detecting a malignant formation at an early stage will be higher and, as a consequence, a decrease in cancer mortality can be provided.

  • We added in the text the following information ( Lines 254 -272):

The ratios of 4 and 1 periods of thermal cycling of all 14 sensors from the heating-cooling point to the cooling-heating point (5.5 seconds * 30 Hz = 165 values) were fed to the input of the neural network. In the course of work, it was experimentally found, if we take every 10 value, then the error does not increase, thus, the size of the input data array significantly reduces (from 165 values ​​for each sensor to 16). Consequently, 16 * 14 values ​​obtained from gas sensors, such as age, sex and the fact of smoking, were fed to the input of the neural network. The total dimension of the input layer was 227 values. The hidden layer has dimension input * 2 or 454 values. The output layer has one neuron that receives "0" in the absence of cancer and "1" if it is present. The transfer function of sigmoid neurons was used on all layers of the neural network.

The error in solving the neural network classification problem was calculated by cross-validation. 59 datasets were randomly divided into 5 groups. In 4 groups were 12 sets, in the 5th - 11. Then training was carried out in 4 groups, using a test set that did not participate in training, based on the fifth group. As a result, 5 experiments were obtained, for each of which the parameters of the informativeness of the investigated diagnostic method were calculated according to the methods [17, 41]. The division into groups was randomly performed 20 times, due to the small number of data sets. For each group five experiments were carried out by it as described above. Table 3 shows the average values ​​for 100 experiments (20 * 5) for the developed gas analytical system.

14 Discussion

Conclusions are not supported due to insufficient e-nose data to demonstrate the basis of conclusions.

Reply:  We added the following information in the text:

  • Lines 225 -240:

Collecting data on a gas sample includes several stages:

  1. A half-filled sample bag is connected to the inlet valve. The bag is loaded with the same weight for all measurements. At this moment, the inlet (2) and outlet (3) valves of the sampling chamber are closed.
  2. In the interface of the data collection program, a button is pressed and the reading and transmission of data at a frequency of 30 Hz to a personal computer begins.
  3. At the moment of transition from the heating phase to the cooling phase of the sensors occurs (Figure 3 - mark 5000 ms), valves (2) and (3) open for exactly one second. The same opening time of valves (2) and (3), and the weight of the load on the bag provide the same volume of gas sample (~ 250 ml) introduced into the 1 liter sampling chamber.
  4. After the valves are closed, data collection continues with the sample gas inside the chamber. The process of collecting data with a gas sample continues until the time stamp of 90,000 ms. The total residence time of the sample in the chamber does not exceed 90 seconds.
  5. After 90 seconds, the sample chamber purge is automatically turned on for 120 seconds. The instrument is then ready to analyze the next sample. These time intervals were selected experimentally and take into account the inertia of the sensors.

  • Lines 245 -246:

As a result, a record of 43200 values is saved in the database for each individual patient (14 sensors and 2 channels, each 30 Hz * 90 seconds = 2700).

15 References

No publication year was provided for references 6, 7, and 8.

Reply:  Corrected

Line 348: We added year of publication for reference N 6

Line 350: added year of publication for reference N 7.

Line 353: added year of publication for reference N 8.

Reviewer 2 Report

The authors used a self-assembled electronic nose with an array of MOS sensors, to classify patients by breath samples analysis into sick and healthy. Digitization and preprocessing of the data were analyzed by a neural network with perceptron architecture.

The work is well written and drawn up in the groove of numerous similar researches in the literature.

the novelty of the work rely in the self-assembled device as the diagnosys itself has been extensively studied in literature

line 170: please specify the abbreviations EVON and VLDPE and make a justification for this choice by describing their characteristics and differences with respect to Tedlar for example. 

line 185: the re is no data about the volume of the sampling chamber, the flow and residence time of the air sample during the measure. Please add those parameters

line 207: this phrase is meaningful in the context please check: "Regardless, the pump continuously blew through the chamber"

line 212-229: the authors set the data sampling time at 90s and a sampling rate of 30 Hz: it is not clear the array of data what dimensions has: this as practical implications in the predictors/samples computed ratio.

It seems at first glance that the number of predictors (signals of the sensors) is very high with respect to samples (people). This very high ratio may increase the chance correlation and/or overfitting of the classification method. The authors should exclude this possibility in a convincing manner in the text. See also: 

Author Response

Reply to Reviewer #2

We are thankful to Reviewer for attention to our work, for such detailed consideration, useful comments, additions and corrections. All the corrections suggested by Reviewer 2 were made over the text of manuscript.

1 Line 170: please specify the abbreviations EVON and VLDPE and make a justification for this choice by describing their characteristics and differences with respect to Tedlar for example. 

Reply: Corrected.

Line 182: We added the text: “Ethylene-Vinyl alcohol (EVOH)”.

Line 183: We added the text: “Very Low-Density Polyethylene (VLDPE)”.

The bags, that we have selected, have some advantages over Tedlar bags. The characteristics of the Tedlar material (PVF) are shown in Figure 1, that also demonstrates the characteristics of the outer layer of our bags (EVOH). EVOH is superior to Tedlar in Permeability coefficient (P0) of oxygen. The inner layer of the VLDPE bags outperforms Tedlar bags in terms of the water vapor transmission rate, which can be compared in Figure 2. Two layers provide superior performance.

Figure 1: page 11 https://www.pnnl.gov/main/publications/external/technical_reports/PNNL-26070.pdf 

Figure 2: page 563 https://books.google.ru/books?id=A-NgDwAAQBAJ 

2 Line 185: there is no data about the volume of the sampling chamber, the flow and residence time of the air sample during the measure. Please add those parameters.

Reply: We added the following information (Lines 225-240):

Collecting data on a gas sample includes several stages:

  1. A half-filled sample bag is connected to the inlet valve. The bag is loaded with the same weight for all measurements. At this moment, the inlet (2) and outlet (3) valves of the sampling chamber are closed.
  2. In the interface of the data collection program, a button is pressed and the reading and transmission of data at a frequency of 30 Hz to a personal computer begins.
  3. At the moment of transition from the heating phase to the cooling phase of the sensors occurs (Figure 3 - mark 5000 ms), valves (2) and (3) open for exactly one second. The same opening time of valves (2) and (3), and the weight of the load on the bag provide the same volume of gas sample (~ 250 ml) introduced into the 1 liter sampling chamber.
  4. After the valves are closed, data collection continues with the sample gas inside the chamber. The process of collecting data with a gas sample continues until the time stamp of 90,000 ms. The total residence time of the sample in the chamber does not exceed 90 seconds.
  5. After 90 seconds, the sample chamber purge is automatically turned on for 120 seconds. The instrument is then ready to analyze the next sample. These time intervals were selected experimentally and take into account the inertia of the sensors.

3 Line 207: this phrase is meaningful in the context please check: "Regardless, the pump continuously blew through the hamber"

Reply: Corrected.

Lines 219-221: While preparing the gas analytical system for measurements, the pump blows clean air through the sampling chamber.

4  Line 212-229: the authors set the data sampling time at 90s and a sampling rate of 30 Hz: it is not clear the array of data what dimensions has: this as practical implications in the predictors/samples computed ratio.

Reply: We added the text:

  • Lines 245-246:

As a result, a record of 43200 values is saved in the database for each individual patient (14 sensors and 2 channels, each 30 Hz * 90 seconds = 2700).

  • Lines 254-263:

The ratios of 4 and 1 periods of thermal cycling of all 14 sensors from the heating-cooling point to the cooling-heating point (5.5 seconds * 30 Hz = 165 values) were fed to the input of the neural network. In the course of work, it was experimentally found, if we take every 10 value, then the error does not increase, thus, the size of the input data array significantly reduces (from 165 values ​​for each sensor to 16). Consequently, 16 * 14 values ​​obtained from gas sensors, such as age, sex and the fact of smoking, were fed to the input of the neural network. The total dimension of the input layer was 227 values. The hidden layer has dimension input * 2 or 454 values. The output layer has one neuron that receives "0" in the absence of cancer and "1" if it is present. The transfer function of sigmoid neurons was used on all layers of the neural network.

The error in solving the neural network classification problem was calculated by cross-validation. 59 datasets were randomly divided into 5 groups. In 4 groups were 12 sets, in the 5th - 11. Then training was carried out in 4 groups, using a test set that did not participate in training, based on the fifth group. As a result, 5 experiments were obtained, for each of which the parameters of the informativeness of the investigated diagnostic method were calculated according to the methods [17, 41]. The division into groups was randomly performed 20 times, due to the small number of data sets. For each group five experiments were carried out by it as described above. Table 3 shows the average values ​​for 100 experiments (20 * 5) for the developed gas analytical system.

5 It seems at first glance that the number of predictors (signals of the sensors) is very high with respect to samples (people). This very high ratio may increase the chance correlation and/or overfitting of the classification method. The authors should exclude this possibility in a convincing manner in the text.

Reply:  We added the text

  • Lines 264-272:

The error in solving the neural network classification problem was calculated by cross-validation. 59 datasets were randomly divided into 5 groups. In 4 groups were 12 sets, in the 5th - 11. Then training was carried out in 4 groups, using a test set that did not participate in training, based on the fifth group. As a result, 5 experiments were obtained, for each of which the parameters of the informativeness of the investigated diagnostic method were calculated according to the methods [17, 41]. The division into groups was randomly performed 20 times, due to the small number of data sets. For each group five experiments were carried out by it as described above. Table 3 shows the average values ​​for 100 experiments (20 * 5) for the developed gas analytical system.

  • Lines 275-279:

Taking into account the small number of data sets, in order to exclude overfitting of the neural network, the control parameter for stopping training was chosen the accuracy in determining the test set. The stop was made at the moment when the error in determining the training sets decreased, but the error in determining the test set began to grow. This point was observed between 500 and 8000 learning epochs.

Round 2

Reviewer 1 Report

The authors have made a very minimal and inadequate attempt to correct clearly identified deficiencies in the manuscript. In particular, the Introduction does not adequately cover the subject matter with appropriate and recent references and the Results section does not display sufficient data to demonstrate the efficacy of the electronic-nose data analyzed or the Conclusions made from this very minimal data.

Reviewer 2 Report

After major revisions in its present form the paper is worth of publication as the authors have welcomed all the advices that I have addressed to.